# Shale gas reserve evaluation by laboratory pyrolysis and gas holding capacity consistent with field data

Patrick Whitelaw[1,2], Clement N. Uguna[1,2], Lee A. Stevens[1], Will Meredith [1], Colin E. Snape [1,2], Christopher H. Vane [2], Vicky Moss-Hayes[2] & Andrew D. Carr[3]

Exploration for shale gas occurs in onshore basins, with two approaches used to predict the maximum gas in place (GIP) in the absence of production data. The first estimates adsorbed plus free gas held within pore space, and the second measures gas yields from laboratory pyrolysis experiments on core samples. Here we show the use of sequential high-pressure water pyrolysis (HPWP) to replicate petroleum generation and expulsion in uplifted onshore basins. Compared to anhydrous pyrolysis where oil expulsion is limited, gas yields are much lower, and the gas at high maturity is dry, consistent with actual shales. Gas yields from HPWP of UK Bowland Shales are comparable with those from degassed cores, with the ca. 1% porosity sufficient to accommodate the gas generated. Extrapolating our findings to the whole Bowland Shale, the maximum GIP equate to potentially economically recoverable reserves of less than 10 years of current UK gas consumption.

[1] University of Nottingham, Faculty of Engineering, Energy Technologies Building, Triumph Road, Nottingham NG7 2TU, UK. [2] British Geological Survey, Centre for Environmental Geochemistry, Keyworth, Nottingham NG12 5GG, UK. [3] Advanced Geochemical Systems Ltd., 1 Towles Fields, Burton-on-the-Wolds, Leicestershire LE12 5TD, UK. Correspondence and requests for materials should be addressed to C.E.S. (email: colin.snape@nottingham.ac.uk)

Shale gas arises from the cracking of insoluble organic matter in source rocks (kerogen) and any oil retained in the pores[1–3]. Shale gas produced in the USA is generally quite dry with methane contents being typically over 75%[3–5] with shales needing a vitrinite reflectance (VR) maturity of >1.4% Ro to produce dry gas[6]. To guide exploration and development where production has not commenced, it is essential that rigorous methodologies are established to estimate the maximum recoverable reserves. The UK is such a case, with the Carboniferous Bowland-Hodder Shale being the major gas source[7–11]. It has been estimated that the gas in place (GIP) for the entire Bowland Shale is large, the Upper and Lower units containing 164–447 and 658–1834 trillion standard cubic feet (TCF), respectively[7]. However, this was based on adsorbed and free gas estimates for US shales, and assumed that all Bowland Shale source rock with a maturity above 1.1% Ro, had already generated gas, contrary to US producing shales (Barnett, Marcellus and Fayetteville) having VR >1.4% Ro[6]. The large UK estimate may also be due to the assumption that all Carboniferous Shales of the Bowland basin are potential shale gas source rocks[7].

Rock-Eval pyrolysis is the standard approach for assessing source rock potential and quality in which volatile hydrocarbons are measured as they evolve[12,13]. Although hydrocarbon gases are not measured, an empirical relationship based on the S1 and S2 parameters (free and potential for generated hydrocarbons, respectively) to estimate shale gas yields has been developed[6]. Closed system pyrolysis uses micro-scale sealed-vessels (MSSV) where all volatiles are retained within the system[14,15]. The drawback with both techniques is that they do not replicate oil expulsion during maturation. In hydrous pyrolysis, albeit in a closed system, oil generated is expelled into the water phase and is thus not in as close contact with the source rock, so better replicating actual expulsion[16]. However, water and vapour are in equilibrium, with the pressure set by the temperature of the experiment. To better replicate petroleum systems, high pressure water pyrolysis (HPWP), where there is no free vapour space in the reactor can be used to understand source rock maturation, hydrocarbon generation and associated pressure effects[17–20].

We use sequential HPWP here to predict the maximum GIP using oil window and gas window mature UK Bowland Shales with expelled oil being removed at each stage. Comparisons are drawn firstly with recent reports for degassed core samples[21,22] and then the adsorbed plus pore (free) gas estimated for the gas window shale. It must be remembered that some differences between the different studies arise from the samples coming from different locations within the basin with consequent differences in sediment provenance, stratigraphical, structural and tectonic histories of the different parts of the same basin[12]. Moisture equilibration is essential since it affects both the free and adsorbed gas, and vast reductions in the amount of adsorbed methane with increasing humidity have been reported[23]. Further, much of the variation in the reported porosities of shales (1–8%) arises from the extent to which shales are moisture-equilibrated[6,24,25]. The implications of our findings for the entire Bowland Shale gas resource are considered on the basin and we show that these are actually ~10 times lower than previously thought.

## Results

**Gas and oil yields.** The methane and total hydrocarbon ($C_1$–$C_5$) gas yields from the five stages in sequential HPWP for the oil window mature Rempstone shale (0.71% Ro, containing a mixture of types II, III and IV kerogen, Supplementary Table 1, Supplementary Fig. 1) investigated at 800 bar and under anhydrous conditions are presented in Figs. 1a, b, respectively,

together with the yields of oil expelled and the heavier oil/bitumen retained in the shale. The full gas compositions, vitrinite reflectance and Rock-Eval pyrolysis results for the matured shale samples from these experiments are listed in Supplementary Table 2, together with those for experiments at 300 bar. Maturities >2.3% Ro were attained to represent the high maturities of the gas window. Slightly higher Ro values were achieved at 300 bar due to the previously described pressure retardation effect on maturation at higher pressure[20].

In HPWP, oil generation peaks at VR of 1.0% Ro (stage 1), and extremely dry gas generation at >2% Ro only commences when the residual oil level is reduced to ca. 5% of its maximum value at the end of stage 2 (Fig. 1a), corresponding to only 1% of the initial total organic carbon (TOC) of Rempstone shale. However, extracting the residual oil after stage 2 (1.3% Ro) to effectively increase the extent of expulsion to over 90% reduced the gas yield by nearly 50% from ca. 22 to 11 mg (g TOC)$^{-1}$ and increased the dryness to over 60% (Fig. 1a), with the dry gas yield from stages 4 plus 5 at >2% Ro being similar (12 and 14 mg (g TOC)$^{-1}$) for both the 800 bar unextracted and extracted Rempstone shale (Fig. 1a, Supplementary Table 2). The small quantities of retained oil present after stage 3 contributed with the higher maturity to make the gas from stage 4 considerably dryer. The gas yields obtained at 800 bar were very similar to the 300 bar yields for Rempstone (Fig. 2 and Supplementary Table 2) confirming that over the maturity range of 1.3–2.0% Ro, the retained oil levels dictate the amount of gas generated, with the dryness increasing with decreasing gas yield. In geological settings before uplift occurs, it is likely that nearly all the oil will be expelled based on the evidence for US shales, for example, Marcellus, where relatively dry gas is obtained at >1.4% Ro[6].

Overall, the gas yields from the HPWP experiments are nearly three times lower than from anhydrous pyrolysis, with the gas under anhydrous conditions being considerably wetter at high maturities (>2% Ro) with a dryness of only 66 and 69% (stages 4 and 5, respectively, Fig. 1b). This vast difference arises because virtually no expulsion of oil occurs in anhydrous pyrolysis with the retained oil/bitumen remaining constant after stage 2 compared with 80% expulsion occurring in HPWP. Cracking of oil gives considerably wetter gas than direct generation from kerogen[20]. Our total gas yield from anhydrous pyrolysis (137 mg (g TOC)$^{-1}$) is comparable to that reported in a previous study (mean of 154 mg (g TOC)$^{-1}$ for source rocks with hydrogen indexes (HIs) of ca. 400[14].

Figure 3 presents the gas and the expelled and retained oil/bitumen yields for the gas window (Grange Hill core, 1.95% Ro), where the HPWP experiment commenced at stage 3 due to its high starting maturity. The full gas compositions, vitrinite reflectance and Rock-Eval pyrolysis results for the matured shales are listed in Supplementary Table 3. Relatively dry gas has been obtained from the first stage (stage 3) of the experiment between 1.9 and 2.1% Ro and the dryness then increases to over 90% for stages 4 and 5. Due to the shorter time spent at higher temperature in stage 3 for this maturity range, dry gas generation has been brought forward. The stage 3 gas yield for Grange Hill is much lower than for Rempstone (Figs. 1a, b and 2) due to the higher starting maturity. The low yield of gas from stage 5 (2.3–2.5% Ro) indicates that the end of the gas window has been reached. The dry gas yields at maturities greater than ca. 2% Ro are similar for both cores (10–14 mg (g TOC)$^{-1}$) indicating that any variations in kerogen type do not impact significantly on gas yields at high maturities.

**Comparison with degassed cores.** Assuming that all the gas generated remains in the shale, converting the HPWP gas yields

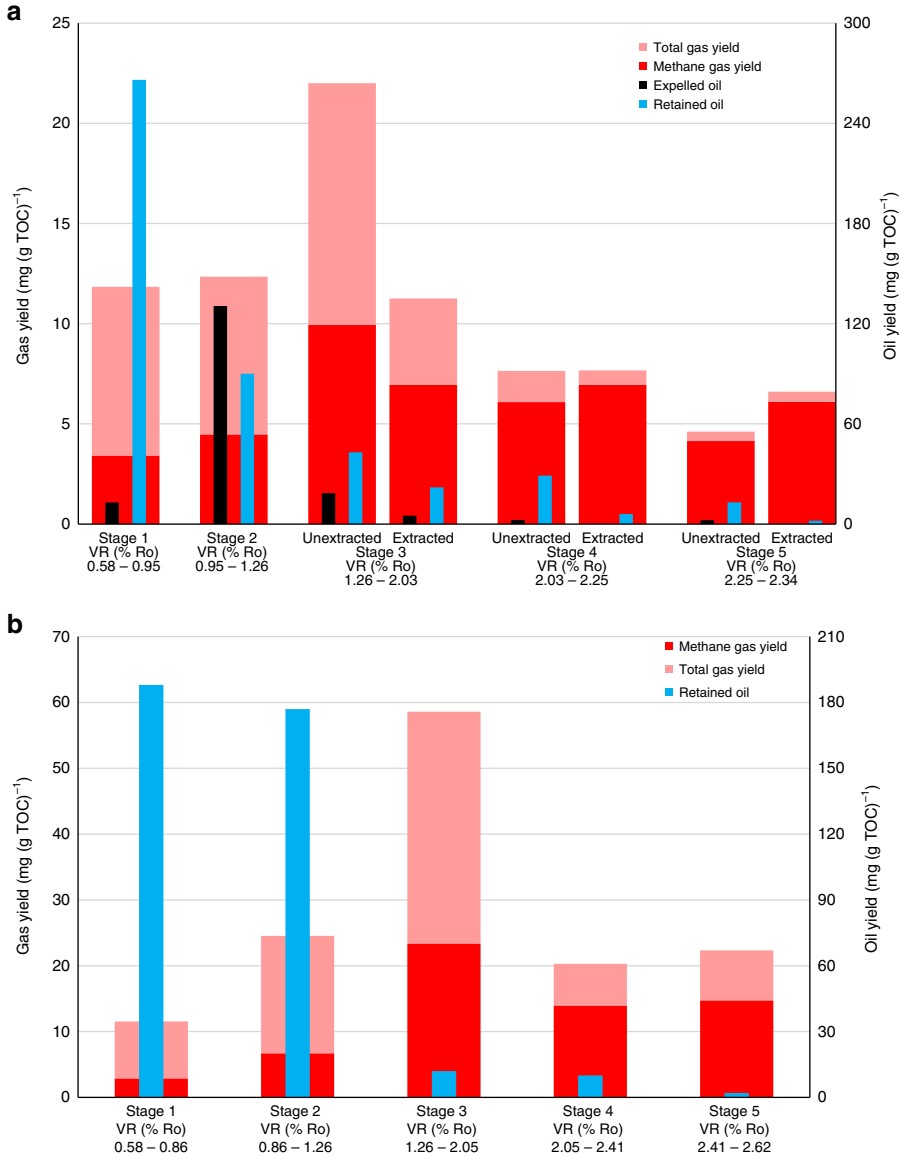

**Fig. 1** Total hydrocarbon gas (C$_1$–C$_5$), methane, expelled oil and retained oil/bitumen yields (mg/g TOC of the rock at the end of each stage) for the Rempstone shale. **a** Sequential HPWP (800 bar). **b** Anhydrous experiments. For stages 3–5 in the HPWP experiment, the results are also presented for the sample that was solvent extracted at the end of stage 2. The VR % (Ro) values below each histogram are for the start and end of each stage. The differences in the measured values from duplicate tests are generally within 6% for gas (C$_1$–C$_5$ hydrocarbons) yields, 4% for expelled and retained oil yields and 2% for gas dryness

for stages 3–5 for Rempstone shales to a volumetric basis normalised to a TOC content of 2% (the mean for the whole Upper Bowland Shale) gives a total of 22–28 TCF tonne$^{-1}$ with 8–14 TCF tonne$^{-1}$ being generated over the range 1.3–2.0% Ro, the uncertainty arising due to the profound effects of the relatively small amounts of retained oil/bitumen have on the gas yields. The desorbed gas content (adsorbed gas measured from desorption experiment) obtained for Grange Hill and two neighbouring wells were in the range 20–50 TCF tonne$^{-1}$ at 1.9–2.3% Ro[22], generally increasing with maturity, mean values normalised to a TOC of 2.0% being 25–28 TCF tonne$^{-1}$ for the Lower Bowland Shales investigated. A range of 10–50 TCF tonne$^{-1}$ has been reported for the Kirby Misperton-8 well in the Cleveland Basin covering a maturity range of 1.3–2.0% Ro[21], but mainly at the higher maturities, where normalising to a TOC of 2.0% gives a mean of 12 TCF tonne$^{-1}$. Overall, to achieve this level of consistency between the HPWP results and the degassed cores

implies that most of the gas generated from 1.3% Ro is retained in the shales.

**Porosity and adsorbed gas measurements.** The data for Nitrogen (N$_2$) sorption isotherms, mercury intrusion porosimetry (MIP) and X-ray computer tomography (XRCT) for the gas window Grange Hill core are presented in Figs. 4 and 5. Figure 4 shows that the Grange Hill initial to stage 5 samples have a type IV isotherm representing a meso and macroporous pore network with mesopore volume increasing with maturity during HPWP (Supplementary Table 4). However, at 50% relative humidity (RH), the mesopore volume observed by N$_2$ adsorption isotherms decreases by 35–40% for Grange Hill initial and 39% for stage 5. Maturation in HPWP (stage 5) did not induce a significant change in the macropore volume (Fig. 5) and the increases in Brunauer-Emmett-Teller (BET) surface area and pore volumes

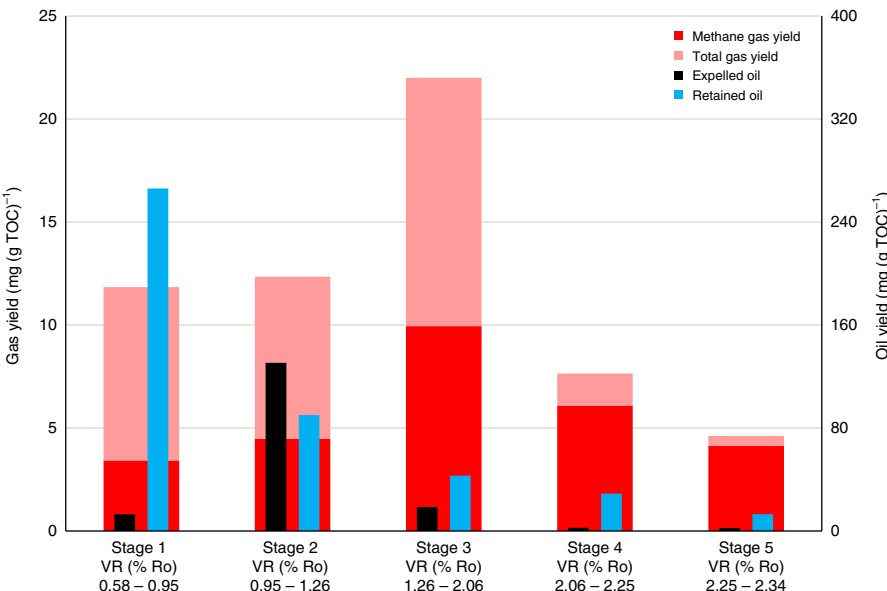

**Fig. 2** Total hydrocarbon gas (C$_1$–C$_5$), methane, expelled oil and retained oil/bitumen yields (mg/g TOC of the rock at the end of each stage) for the HPWP experiment on Rempstone shale at 300 bar. The VR % (Ro) values below each histogram are for the start and end of each stage. The differences in the measured values from duplicate tests are generally within 6% for gas (C$_1$–C$_5$ hydrocarbons) yields, 4% for expelled and retained oil yields and 2% for gas dryness

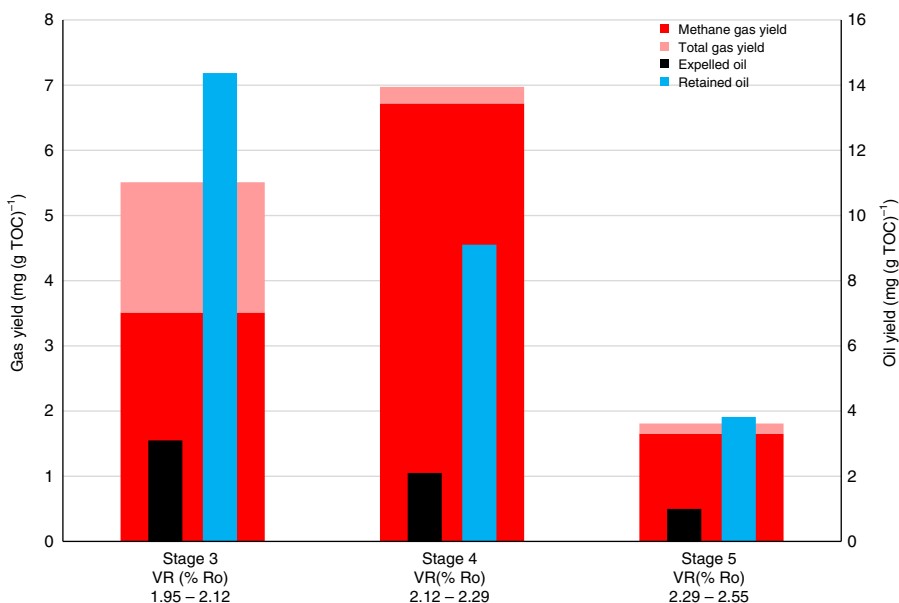

**Fig. 3** Total hydrocarbon gas (C$_1$–C$_5$), methane, expelled oil and retained oil/bitumen yields (mg/g TOC of rock at the end of each stage) for the sequential HPWP experiment on the Grange Hill core sample at 300 bar. The VR % (Ro) values below each histogram are for the start and end of each stage. The differences in the measured values from duplicate tests are generally within 6% for gas (C$_1$–C$_5$ hydrocarbons) yields, 4% for expelled and retained oil yields and 2% for gas dryness

during the gas window (Supplementary Tables 4 and 5) are broadly consistent with those reported in a previous study[26]. MIP can only be conducted on vacuumed dry samples, but still shows mesopores are dominant. Mesopore volumes from MIP are a factor of 6 greater than for N$_2$ adsorption (Supplementary Tables 4 and 5), this can be attributed to a 'pore shielding' mechanism, where mercury is shielded from a large cavity by a narrower neck/window size pore in the mesopore range, once intrusion occurs the large cavity volume is added to the mesopore volume. This is also evident in N$_2$ isotherms, which show (Fig. 4) nitrogen condensate can only desorb from larger mesopores when

a narrower neck empties, indicating possible ink-bottle pores from the hysteresis in the desorption branch. MIP indicates that the dry porosity is only 1.1% for the initial shale (Supplementary Table 4, multiplying 0.0042 cm$^3$ g$^{-1}$ by the skeletal density of 2.689 g cm$^{-3}$ from helium pycnometry). However, this low value could be partially attributable to the drilling mud present. After HPWP, the dry porosity increases to ca. 7% but, with a moisture content of ca. 2% w/w, corresponding to ca. 5% by volume, means that the wet porosity will be close to 1%.

Macropores imaged by XRCT are shown in Fig. 5, and volume and size distributions listed in the Supplementary Table 6. The

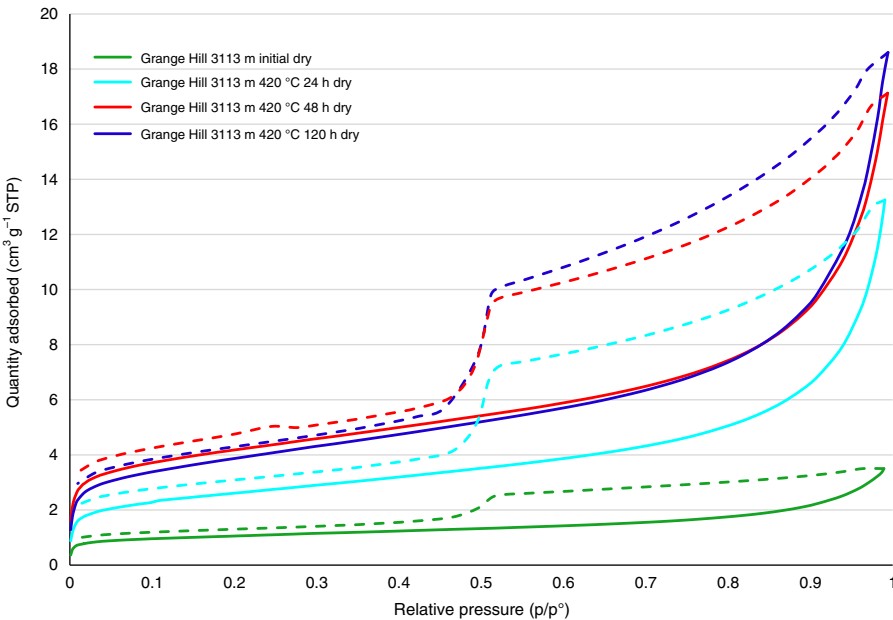

**Fig. 4** $N_2$ sorption isotherms for dry initial and matured Grange Hill shale samples from sequential HPWP pyrolysis (solid lines are for adsorption and dotted lines for desorption)

large fissures induced by HPWP have created an additional 1.4% porosity for the moisture-equilibrated sample used for XRCT. Taking the XRCT pore volume for the initial shale and adding this to the meso and micropore volume from $N_2$ adsorption for the shale equilibrated at 50% RH, gives a total porosity little more than the 0.4% observed by XRCT. The low $N_2$ adsorption meso and micropore volumes could be influenced by the drilling mud present. However, even taking the dry porosity for the drilling mud extracted sample, which will be an over-estimate, gives a micro/mesopore porosity of ca. 0.6% and a total close to 1.0%. For the HPWP stage 5 sample, this analysis gives a porosity of 1.6% for the 50% RH data from $N_2$ adsorption isotherms, which will be lower at 100% RH. Further, it is uncertain whether the HPWP treatment in itself increases micro/mesopore volume, given that a small increase was observed by XRCT for the 2.75–40 μm macropores, but this is not expected to be significant at high humidity. Either way, the evidence overall suggests that 1.0% is a reasonable estimate of the porosity at high humidity for the Grange Hill shale.

The high-pressure methane adsorption isotherms obtained for the HPWP pyrolysis matured gas window Grange Hill shale, both dry and moisture equilibrated (50 and 100% RH) at 25, 60 and 100 °C (Fig. 6), all display type 1 isotherms indicating micropore filling behaviour. Supplementary Table 7 lists the adsorption capacities at 100 and 300 bar, including monolayer capacities derived using duel site Langmuir equation. The stage 5 sample, dry at 25 °C, shows the largest methane adsorption with a monolayer capacity ($Q_m$) of 1.37 mg g$^{-1}$. Overall, the results confirm adsorption capacities increase with maturity[25,26]. Micropores are reduced considerably after equilibration with moisture at 50% RH, $Q_m$ dropping by 27% to 1.00 mg g$^{-1}$ for the stage 5 sample. However, this reduces further when taking into account both the temperature and the assumed humidity at the depth of this shale, 100 °C and 100% RH, respectively, reducing adsorption further by another 85% to a $Q_m$ of 0.15 mg g$^{-1}$. Thus, the combined effect of humidity (dry to 100% RH) and temperature going from ambient to 100 °C is to reduce the equilibrium methane adsorption capacity by a factor of 9 consistent with previous studies[23,27]. The amount of adsorbed methane is reduced further if present-day pressures of shales are below ca.

350 bar where equilibrium is reached (Fig. 6). On the other hand, these estimates may be low if capillary condensation is neglected[28], but this only occurs to a significant extent for wet gas. Methane adsorption capacities reported for other shales range from 0.26 (Eagle Ford) to 1.50 mg g$^{-1}$ (Barnett) for US shales (measured at 40–50 bar)[29] and between 1.00 and 4.08 mg g$^{-1}$ for Qiongzhusi shale, China (measured at 140 bar)[30]. Not surprisingly, these estimates are considerably lower than for isolated type II kerogen[31], which had an adsorption capacity of 15 mg g$^{-1}$.

## Discussion

To compare the GIP estimates from our pyrolysis experiments and the adsorbed plus free gas measurements for the Grange Hill core, the present-day temperature and hydrostatic pressure of 100 °C/300 bar matching many gas window Bowland Shales with ca. >2.0% Ro (Supplementary Fig. 2). Figure 7 compares the adsorbed and free gas estimates for this scenario for shales with the HPWP gas, assuming a porosity of 1% for the water equilibrated shale. The fact that the maximum HPWP yield of 37 TCF tonne$^{-1}$ for Rempstone adjusted to a TOC of 3.0%, to match the value for the Grange Hill core, across the whole gas window is comparable to the shale holding capacity indicating that overpressure will only occur either at higher TOCs or if the porosity is much less than 1%.

To extrapolate our findings to estimate the maximum GIP, the calculation procedure described in the Methods section was used. Our estimates have been calculated apportioning the estimated mean net Upper Bowland Shale volume (32.9 TCF) used previously[7] between the thermal maturity ranges studied by HPWP for the gas window. Thus, we estimate that the shale volume in the gas generating window (>1.3% Ro) is probably only 21.5 ± 3 TCF with 8 ± 2 TCF at >2.0% Ro. Note that this estimate of the Bowland Shale volumes in the particular maturity ranges take no account of whether or not the rock formation is currently at depths >1500 m, which is the base for gas-shale production[7]. The gas yield for the unextracted Rempstone shale from stages 3, 4 and 5 was assumed to be the upper bound of gas generation, while the corresponding yield for the extracted sample, the lower

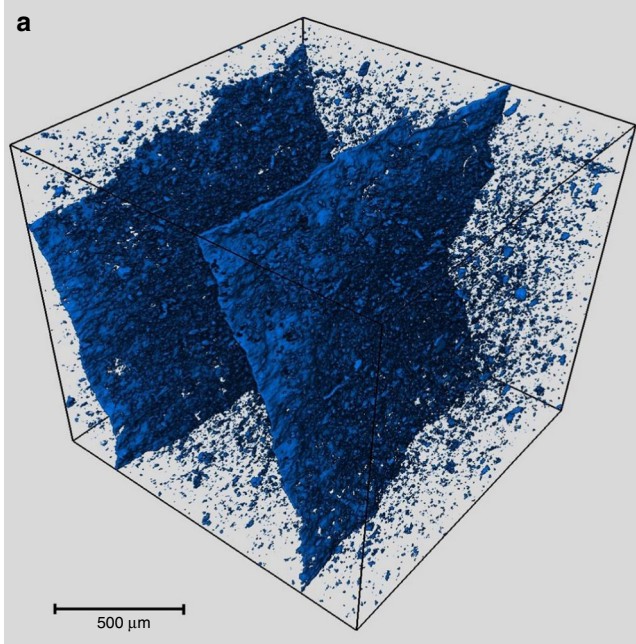

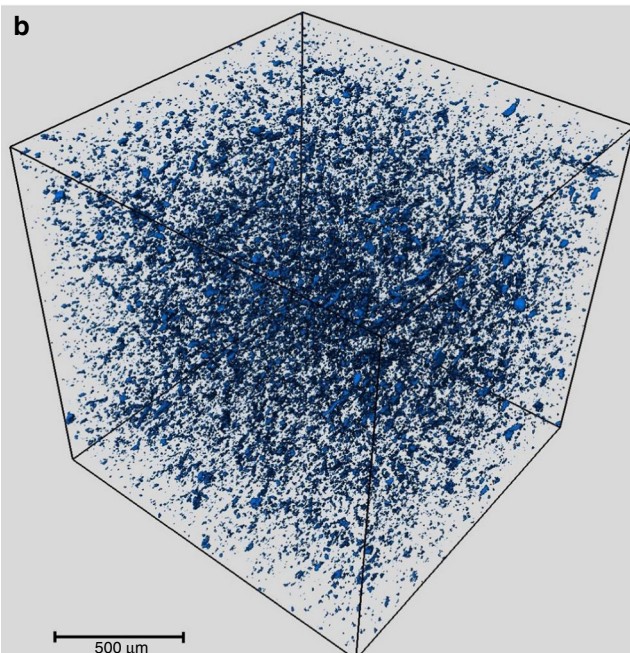

**Fig. 5** XRCT pore visualisation for Grange Hill shale sample from stage 5 of the sequential HPWP experiment. **a** All pores showing the fissures induced. **b** Between 2.75 and 40 μm, excluding the larger fissures for comparison with MIP pore range

bound (Supplementary Table 8). To provide a maximum estimate, we assume that the shale in the maturity range 1.3–2.0% Ro has generated the maximum stage 3 yield in HPWP and that at >2% Ro, the total HPWP gas yield for stages 3–5. This gives an estimated total GIP of 28 ± 11 TCF with 16 ± 6 TCF at maturities >2% Ro. This total is ca. 10 times lower than the previous mean estimate[7], a factor of 5 arising from the estimated lower gas yields and a further factor of 2 from tailoring the volume of shale to the maturity range over which relatively dry gas is actually generated.

The Lower Bowland Shale is estimated to be four times larger by volume than Upper Bowland where we assume that the lower average TOC[7] is roughly offset by the higher overall maturity

range arising from its overall greater depth. This takes the maximum GIP estimate to 140 ± 55 TCF. Given that UK gas consumption is currently ca. 2.8 TCF per annum[32] and, assuming an economic recovery of 10%, which is unlikely for much of the Lower Bowland Shale due to its depth of over 3000 m, represents a maximum (14 ± 6 TCF), considerably below 10 years supply at the current consumption. Clearly, more shales need to be investigated covering different lithologies and over smaller maturity increments, particularly in the range 1.3–2.0% Ro, to provide more precise information as to how much lower the actual GIP is than this maximum estimate.

## Methods

**Bowland Shale samples**. The Carboniferous, basal Namurian, Upper Bowland Shale Formation was deposited in parts of the East Midlands, North Wales and Northern England in a series of subsiding grabens and half-grabens[33–36]. The thickness of the entire Bowland-Hodder Shale varies from 3.5 km in parts of East Midlands to ~0.1 km in parts of the Derbyshire Dome and Cheshire Basin (Fig. 18 in ref. [7]), with the most prospective areas being the Bowland Basin (including Fylde), Gainsborough Trough and Widmerpool Gulf and Cleveland Basin (North Yorkshire). The shale contains hemipelagic mudstones and mass-flow limestones, sandstones, and rare volcanics passing laterally into platform/ramp carbonates[7], with these lithologies presumably having lower potential for gas generation than the hemipelagic mudstones. However, these non-shale lithologies within the shales form an essential component of the shales to form a gas-shale source-reservoir rock, since the production of gas via fracking from shales requires that the total clay content is <35%. The proportions of high potential gas source shale and the other lithologies with low potential can be as high as 75% in the Lower Bowland Shale Fylde area (Lancashire) reducing to close to zero in the East Midlands Shelf, although in the nearby Widmerpool Gulf organic-rich hemipelagic shales occur[7]. In the Upper Bowland Shale, shale is the dominant lithology. The estimates for the gas volumes in place assume the presence of 30% shale in the lower part and 50% in the upper part. Although data are sparse, this indicates that the Lower Bowland Shale will have a lower average TOC. However, we have taken the average TOC for the whole shale to be 2.0%[7] to estimate the GIP.

In the Widmerpool Trough, the Remstone-1 well is on the southern edge and the Bowland Shale is underlain by the Widmerpool Formation and other Visean shales, limestone and siltstones. This oil window mature shale rock is from a borehole core (Rempstone-1 well) of Namurian (Pendleian) age obtained at a depth of between 665 and 667 m. Whereas the Grange Hill-1 3113 m sample is from the Lower Bowland Shale (Brigantian, Dinantian) with a provenance from the prodelta sources to the north east, the Rempstone sample comes from the Upper Bowland Shale (Pendleian, Namurian) with a province from the prodeltas to the north and south of the Widmerpool Trough on the Derbyshire and Midlands Highs. Drilling ceased within the Lower Bowland Shale in the Grange Hill-1 well, but the evidence from the nearby Becconsall-1z well and the Clitheroe and Lancaster Fells districts is that the Lower Bowland Shales are underlain by shales and limestones as in the Rempstone-1 well[22]. Cessation of rifting occurred across large parts of the UK during late Visean and was followed by a period of regional thermal subsidence. While shale deposition continued in the Widmerpool Trough until Kinderscountian/Marsdenian times, culminating with the siliciclastic sandstones of the Millstone Grit Group, the Upper Bowland Shale in Grange Hill-1 is overlain by the Pendle Grit (part of the Millstone Grit Group). These sandstones represent the progradation of deltas across the Visean and early Namurian basins. Both the Grange Hill-1 and Rempstone-1 wells were inverted and eroded during the Variscan orogeny during Late Carboniferous prior to deposition of Permo-Triassic rocks. Both these cores have TOCs higher than the average of 2.0% for the Bowland Shale[22].

**Soxhlet extraction to determine bitumen/oil content of rocks**. Soxhlet extraction was used to determine the bitumen/oil content of the core samples and was carried out using a cellulose extraction thimble and a 250 ml round bottom flask. Prior to extraction, the cellulose extraction thimble was pre-extracted using 150 ml dichloromethane (DCM)/methanol mixture (93:7 volume:volume for 24 h to remove any impurities present. The rock sample was ground into a fine powder and placed within the cleaned thimble, and extracted in the same manner as the thimble was cleaned. The extracted sample was then stored for analysis, and the solvent was evaporated using a rotary evaporator until the majority of the solvent was removed. The oil/bitumen remaining after evaporation was transferred to a pre-weighed vial using DCM and left to dry. The weight of the vial and extract was taken and oil/bitumen weight calculated by difference after all the DCM had evaporated.

**Pyrolysis and product analysis**. Prior to pyrolysis, the non-extracted cores were crushed to 2–5 mm chips that were thoroughly mixed to obtain a homogenous sample. Sequential pyrolysis tests have been carried out under anhydrous (5–20 bar) and high-pressure water (300 and 800 bar) conditions in a 25 ml Hastalloy

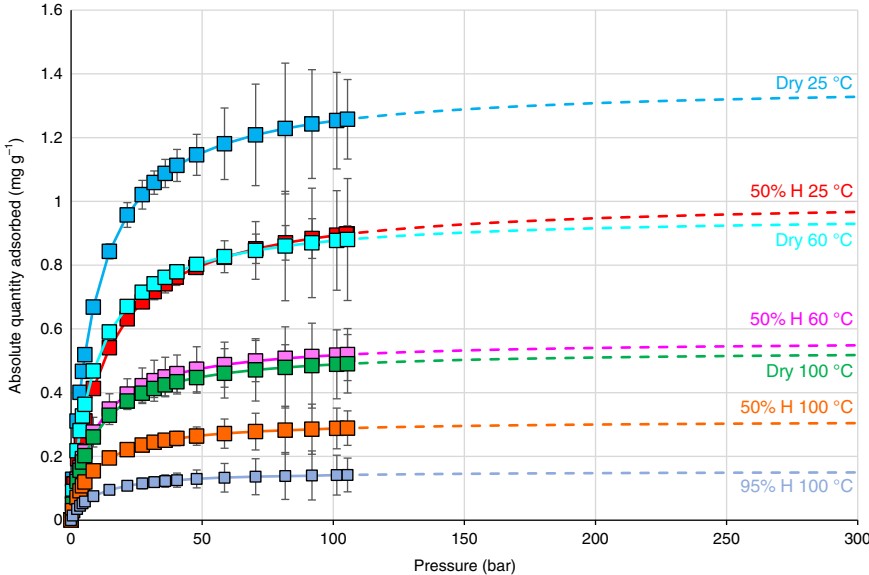

**Fig. 6** High-pressure methane adsorption isotherms fitted to the dual site Langmuir model (dashed line) for 50% RH, 100% RH and dry samples at 25, 60 and 100 °C for matured Grange Hill shale samples from step 5 of the sequential HPWP experiments. The data points are the mean from duplicate experiments and the error bars represent the difference between each pair of values obtained

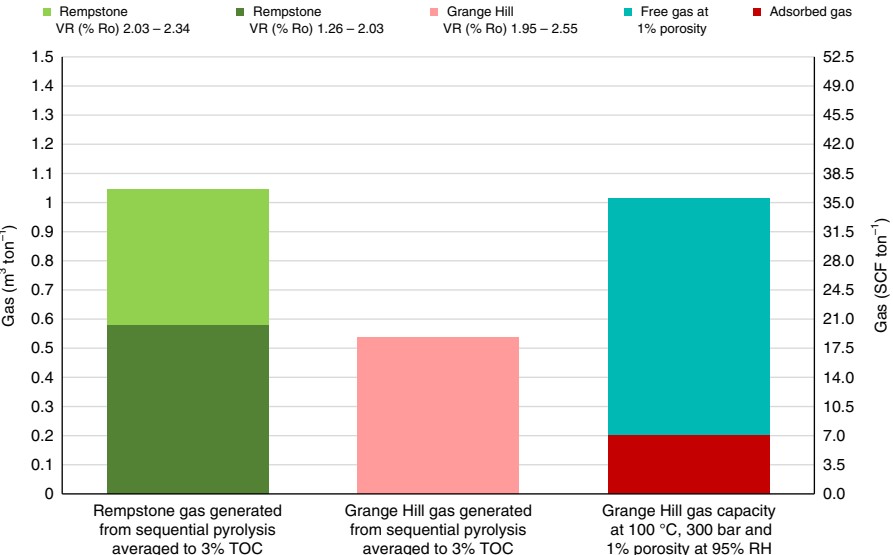

**Fig. 7** Comparison of gas generated at %Ro >1.3 and shale holding capacity (free plus adsorbed gas) based on the results for Rempstone and Grange Hill shales normalised to 3% TOC (TOC after stage 5 of HPWP for Grange Hill)

cylindrical pressure vessel rated to 1400 bar at 420 °C, connected to a pressure gauge and rupture disc rated to 950 bar. Heat was applied by means of a fluidised sand bath, controlled by an external temperature controller. The sand bath (connected to a compressed air source) was pre-heated to the required experimental temperature and left to equilibrate before the start of each run. For all experiments, after the addition of sample and water (for runs with water added) to the rector and reactor assembly, the reaction vessel system was flushed with nitrogen gas to replace air in the reactor head space. After which 2 bar pressure of nitrogen was pumped into the pressure vessel system to produce an inert atmosphere during the pyrolysis runs.

The 300 bar experiments at 350 and 380 °C were performed by initially filling the vessel with 15 ml distilled water, after which the pressure vessel was then lowered into the sand bath and allowed to attain vapour pressures of 175 and 235 bar at 350 and 380 °C, respectively, before the addition of excess distilled water via a compressed air driven liquid pump to increase the pressure to 300 bar. The 300 bar run at 420 °C was conducted by adding 10 ml distilled water to the vessel, the expansion of the water gave the required pressure and the experiment was not pressurised. The 800 bar experiment at all temperatures were performed similarly to the 300 bar runs at 350 and 380 °C, also filling the vessel initially with 15 ml distilled water before increasing the pressure to 800 bar. The anhydrous experiment

was also performed in the same manner as the high water pressure runs without water, the 5–20 bar pressure observed was generated due to the expansion of the 2 bar nitrogen in the system during the run and water generated from the shale during the experiment. After the required temperature and pressure for all conditions were attained, the experiments were then allowed to run for the required time, after which the sand bath was switched off and left to cool to ambient temperature before product recovery[17–20].

The sequential experiments were conducted as described above and depicted in Fig. 8. The experiments were conducted starting with 19 g of the oil window mature Rempstone shale for all conditions. The starting rock (0.71% Ro and $T_{max}$ of 441 °C after the removal of suppression of VR[37] and $T_{max}$[38], respectively) was first heated at 350 °C for 24 h, and at the end of the run the experiment was stopped and allowed to cool to ambient temperature, before the generated products (gas, expelled oil and pyrolysed rock) were recovered, and the pyrolysed rock dried to remove water. After drying the rock, about 3 g was put aside for further analysis, and the rest re-heated. The process was repeated heating the same rock sample successively at 380 °C for 24 h, 420 °C for 24 h, 420 °C for 48 h, and finally 420 °C for 120 h.

For the gas mature Grange Hill core with starting vitrinite reflectance of 1.95% Ro, HPWP at 300 bar was conducted in the same manner as the Rempstone

**Fig. 8** Schematic diagram showing the temperatures and times used for the 5 stages in the sequential pyrolysis experiments on the Rempstone shale. The sequential experiment for Grange Hill was started at stage 3 given the initial vitrinite reflectance of 1.95% Ro

sequential pyrolysis, however starting at the third stage of the sequential pyrolysis. The core was heated successively at 420 °C for 24 h, 420 °C for 48 h, and finally 420 °C for 120 h.

The removal of the expelled oil and the gas after each maturity stage enables the maturity interval to be identified over which dry shale gas will be generated. At the higher temperatures of 380 and 420 °C used to reach high maturities, the water is supercritical and could have greater extractive power possibly leading to more oil expelled when compared with 350 °C[19].

After every pyrolysis stage, the gases were collected with the aid of a gas tight syringe and transferred to a gas bag (after the total volume had been recorded), and immediately analysed for the $C_1$–$C_5$ hydrocarbon composition by gas chromatography on a Clarus 580 GC fitted with a FID and TCD detectors operating at 200 °C. Hundred microlitres of gas samples were injected (split ratio 10:1) at 250 °C with separation performed on an alumina plot fused silica 30 m × 0.32 mm × 10 μm column, with helium as the carrier gas. The oven temperature was programmed from 60 °C (13 min hold) to 180 °C (10 min hold) at 10 °C min$^{-1}$. Individual gas yields were determined quantitatively in relation to methane (injected separately) as an external gas standard. The total yield of the hydrocarbon gases generated was calculated using the total volume of generated gas collected in relation to the aliquot volume of gas introduced to the GC, using relative response factors of individual $C_2$–$C_5$ gases to methane predetermined from a standard mixture of $C_1$–$C_5$ gases[19]. The expelled oil floating on top of the water after the experiments was collected with a spatula and recovered by washing with cold DCM (for runs where expelled oil was generated), after which the water in the vessel was decanted and the pyrolysed rock oven dried overnight at 45 °C. The floating (expelled) oil on top of the water, together with oil adhered to the side of the reactor wall (recovered by washing with DCM), were all combined and referred to as expelled oil. About 1 g of the dried pyrolysed rock was crushed and soxhlet extracted as described above to recover the oil retained in the rock (bitumen).

**Vitrinite reflectance (VR)**. Measurements were conducted on the initial (non-extracted) and pyrolysed rocks solvent extracted residues mounted in epoxy resin, using standard methods[39]. Prior to reflectance measurements, the samples were ground and polished using successively finer grades of silicon carbide and colloidal silica to produce a scratch free polish surface. Measurements were made using a LEICA DM4500P microscope with motorised fourfold turret for reflectance. The microscope was fitted with oil immersion objectives. The white light source was a 12 V 100 W halogen lamp with a LED illumination slider 29 × 11.5 mm in the incident light axis. Calibration was carried out using a 3.13% Ro Zirconian standard, and a blank (0% Ro), and was checked using a YAG standard (0.89% Ro) to ensure a linear calibration. Random VR (% Ro) measurements were carried out at 546 nm, and between 6 and 32 points count were taken depending on the number of recognisable vitrinite particles available for measurement in each sample. Measurement and data were collected via the Hilgers Fossil Man system connected to the LEICA DM4500P microscope.

**Rock Eval pyrolysis and total organic carbon (TOC)**. Analysis were conducted on the initial and pyrolysed non-extracted and extracted rocks from the sequential experiments. Rock Eval pyrolysis used a Vinci Technologies Rock Eval 6 standard instrument, with about 60 mg of crushed powdered rock being heated using an initial oven programme of 300 °C for 3 min and then from 300 to 650 °C at the rate of 25 °C min$^{-1}$ in an $N_2$ atmosphere. The oxidation stage was achieved by heating at 300 °C for 1 min and then from 300 to 850 °C at 20 °C min$^{-1}$ and held at 850 °C for 5 min. Hydrocarbons released during the two-stage pyrolysis were measured using a flame ionisation detector (FID) and CO and $CO_2$ measured using an infra-red (IR) cell[20].

**Methane adsorption**. Isotherms were obtained using a Micromeritics High Pressure Volumetric Analyser (HPVA-100) at 25, 60 and 100 °C up to pressures of 100 bar on both moisture-equilibrated and dry shales. The crushed shale samples (2–5 mm) with moisture present (equilibrated at 50 and 100% RH over 48 h) or vacuum dried for 48 h at 80 °C (dry) were loaded into the 10 cm$^3$ sample cell (~10 g). Skeletal densities of the shale were calculated using helium pycnometry on the vacuum dried shale, with the assumption that helium penetrates all accessible porosity. Free space for analysis was calculated by taking the free space of the empty cell calculated from helium expansion minus the volume of the shale. Monolayer capacities (Qm) were calculated using the dual site Langmuir equation to predict adsorption beyond the experimental range as it could not be reached through experimental means[40].

**Table 1 XRCT parameters**

| Parameter | | Parameter | |
| --- | --- | --- | --- |
| Camera temperature | −59 °C | Image size | 2048 × 2048 pixels |
| Source voltage | 80 kV | Pixel size | 2.522 μm |
| Source current | 87 μA | Optical magnification | 0.4× |
| Source filter | LE2 | Exposure time | 5 s |
| Source RA distance | 11–13 mm | Camera readout | 2.5 MHz |
| Detector RA distance | 137–161 mm | Projection number | 1600 |

**N2 sorption isotherms**. BET specific surface area, micro, meso and macroporosity of the shale samples were analysed using a Micrometics ASAP 2420 instrument. Using $N_2$ as the adsorbate at −196 °C, isotherms were acquired from 0.001 to 0.998 relative pressure. About 3 g of shale samples (2–5 mm) were placed into a glass tube with filler rod. Dry samples were vacuumed dried at 80 °C for 15 h prior to analysis. Wet samples (50% RH equilibrated) were frozen at −196 °C in liquid $N_2$ for 30 min in the glass tube with filler rod prior to analysis, with the instrument and sample taken to vacuum manually with frozen water held in the pores and surface of the samples. This method eliminates the free space procedure as the isotherm is started immediately as the vacuum set-point is reached (0.013 mbar), therefore a separate free space analysis was carried out on blank tubes similar to the method above for methane adsorption. Surface areas of the shale were calculated using BET surface area equation from 0.05 to 0.25 relative pressure giving positive BET C values[41]. Micro and mesopore volumes were determined using Horvath-Kawazoe model, assuming slit pore geometry on a carbon/graphite surface.

**Mercury intrusion porosimetry (MIP)**. Macro and mesopore volumes by MIP were measured with a Micrometrics Autopore IV Mercury Porosimeter. The shale (1.5 g, 2–5 mm) was vacuum dried for 48 h at 80 °C, and placed within a 5 cm$^3$ solid penetrometer, 0.366 IV. The pressure was increased stepwise from vacuum up to ~4137 bar and the volume of mercury entering the shale pores can be converted to pore volume and size. The radii of the penetrated pores at a given pressure was calculated using the Washburn equation for slit/angular shaped pores with a contact angle of 151.5° and a surface tension of 475.5 mN/m for mercury intrusion in shale[42] providing a pore size distribution from 231 μm to 3 nm.

**Humidity generation**. Humidity generation was obtained with an oversaturated salt solution placed into a pre-vacuumed desiccator. For 50% RH, 15 g of magnesium nitrate hexahydrate ($Mg(NO_3)_2 \cdot 6H_2O$) was dissolved in 10 ml of distilled water and for 100% RH 8 g of potassium nitrate ($KNO_3$) was dissolved in 10 ml of distilled water[43]. Samples were placed within the desiccator, which was subsequently sealed and evacuated for 3 min. The samples were then left to equilibrate for 48 h at 20 °C.

**X-ray computer tomography**. XRCT measurements were carried out on an Xradia Zeiss Versa XRM500 CT system with a maximum electron acceleration of 160 kV. Images were captured using a 2 × 2 camera binning mode over 180° rotation using parameters in Table 1. Pore size modelling was conducted using the Avizo version 9.0.1 programme. Sub-volumes were extracted using a 600 voxel count per axis equivalent to 1.5 mm. Non-local means filtering was applied using a 21 pixel search window, local neighbourhood of 5 pixels and a similarity value of 0.6. Segmentation was applied to identify pore labelling occurring within the thresholds of 0–5800 for Grange Hill Virgin Extracted and 0–6500 for Grange Hill 300 bar 420 °C 120 h. Volume fraction and labelling was applied identifying pore volume and distribution. Sieve analysis was applied to pores with a diameter between 2.75 and 40 μm with volume fraction and labelling applied to identify pore size volume and distribution within this range to compare with MIP pore range.

**GIP calculation for the entire Bowland Shale**. To estimate the GIP for the entire Bowland Shale from the estimated maturity profile, the individual gas yields were converted from milligram to volume using their different gas densities to obtained the total ($C_1$–$C_5$) gas volume (cubic feet), and the pyrolysed rock converted to volume assuming a bulk shale density[7] of 2.6 g cm$^{-3}$, similar to the Grange Hill core as depicted in Supplementary Fig. 2. Our estimates have been calculated taking the estimated amount of shale in the three different thermal maturity ranges measured in the HPWP experiments, namely, 1.3–2.0, 2.0–2.3 and >2.3% Ro.

The present-day temperature maturity gradients from the petroleum system models[7] were used to assess the maturity range of the Bowland Shale. These were then used to split the percentage volumes of the shale reservoir into various maturity ranges. A hydrostatic gradient was used to predict the pressure-depth histories, as in previous models[7]. The pore pressure is given by the pore fluid density (water assumed), gravitational acceleration and the depth of burial at present day. The advantage of using the same pressure assumptions for assessing the proportions of the Bowland Shale in the different maturity windows is that the maturity-depth gradients in the wells are the same as in the previous reports[7].

The estimates from the area of Bowland Basin at particular levels of maturity were made from the well maturity gradients and present-day depth to the top of the Bowland Shale[7]. As indicated, our estimates have been calculated apportioning the Upper Bowland shale volume using the previously reported median result[7], with a volume of $9.31e11 \text{ m}^3$ (32.9 TCF). The volume of Lower Bowland shale was assumed to be four times that of Upper Bowland[7]. The Basin volume was sub-divided by maturity ranges using the estimations given below.

35% (±10%) between 1.1 and 1.3% Ro;
40% (±15%) between 1.3 and 2.0% Ro;
5% (±2%) between 2.0 and 2.3% Ro;
15% (±5%) between 2.3 and 3% Ro;
5% (±2%) >3% Ro.

## Data availability

The data underlying Figs. 1, 2, 3 & 7 are presented in the Supplementary Tables, and the source data supporting Figs. 4, 5 & 6 are available from the corresponding author upon reasonable request.

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

## Acknowledgements

The authors thank the National Environment Research Council (Grant no: NE/C507002/1 and NE/R018030/1), Statoil and Woodside for supporting the development of the high-pressure water pyrolysis (HPWP) technique. The British Geological Survey, Centre for Environmental Geochemistry for a research fellowship for Dr. Clement Uguna and a studentship for Patrick Whitelaw to conduct the HPWP experimental programme.

## Author contributions

C.E.S. and A.D.C. conceived the project, interpreted the data and wrote the paper. P.W., C.U. and W.M. designed and performed the pyrolysis experiments. P.W. and L.A.S. undertook the gas adsorption, porosity and XRCT measurements. C.H.V. selected the cores and interpreted the data. V.M.H. performed the Rock Eval and TOC analysis.

## Additional information

**Competing interests:** The authors declare no competing interests.

**Peer Review Information:** *Nature Communications* thanks Alastair Fraser and other anonymous reviewers for their contribution to the peer review of this work. Peer reviewer reports are available.

