## [Peer Review File · Nature Communications]

REVIEWERS' COMMENTS:

Reviewer #2 (Remarks to the Author):

I am pleased to see a paper of such high calibre. The authors have also performed diligent proof reading of the manuscript.

This manuscript is highly novel and the choice of pyrolysis and surface area parameters combined with demonstrable numerical understanding of processes such as generation and expulsion come up with a conclusion of major economic significance in that there are recoverable reserves of less than 10% of current gas consumption in the Bowland UK shales.

There is a paper that has been published (Ewbank et al., 1995) which is not cited by the authors that shows the main phases of hydrocarbon generation and mineralization appear to be unrelated in the South Pennine orefield and suggests a late Carboniferous age for hydrocarbon generation. This is best cited with the other references in line 3 under Methods Bowland shale samples together with (Fraser and Gawthorpe, 1990; Leeder, 1982, 1988).

Reference to add:

Ewbank, G., Manning, D.A.C. & Abbott, G.D. 1995. The relationship between bitumens and mineralization in the South Pennine Orefield, central England. *Journal of the Geological Society*, London, 152, 751–765. <https://doi.org/10.1144/gsjgs.152.5.0751>

In conclusion, there are no other papers in the literature that have treated the subject referred to here by the authors, therefore I conclude this is worthy of publication

Reviewer #3 (Remarks to the Author):

I like the paper and the main aim of providing a more accurate (or at least different) method of assessing in place shale gas resources in the subsurface in the absence of long term well test data. It is essentially a static methodology that is particularly useful in new basins where a long history of production (dynamic data) is not available to inform investment decisions

The paper uses two cored Bowland Shale data points from the northern England Carboniferous province namely Grange Hill in the Bowland Basin and Rempstone -1 in the Widmenpoo; Gulf. The boreholes are roughly 200kms apart and have been chosen to compare similar shales at different maturity levels. They are also different ages and this is not discussed. They also have potentially different provenance areas; Grange Hill from the north east and Rempstone from the south. Probably not that relevant to the research but should be mentioned.

My understanding of the maturity of the shales at Rempstone differs somewhat from the authors. My analyses indicate these are early mature for oil generation. In fact it is difficult to explain the conventional Rempstone oilfield interbedded with the Bowland Shale in the absence a mature local source rock. This observation would not make a big difference to the analysis and subsequent conclusions.

The detailed analysis of the methods used could be in my view shortened and replaced by some key information on the resource calculation and estimates currently consigned to supplementary information.

Manuscript entitled “Shale gas reserve evaluation by laboratory pyrolysis and gas holding capacity consistent with field data” by Whitelaw et al.

We thank the reviewers for their supportive and constructive comments. The changes made to this manuscript in response to the Referee’s comments are listed below.

Reviewer 2

Comment: There is a paper that has been published (Ewbank et al., 1995) which is not cited by the authors that shows the main phases of hydrocarbon generation and mineralization appear to be unrelated in the South Pennine orefield and suggests a late Carboniferous age for hydrocarbon generation. This is best cited with the other references in line 3 under Methods Bowland shale samples together with (Fraser and Gawthorpe, 1990; Leeder, 1982, 1988). Reference to add: Ewbank, G., Manning, D.A.C. & Abbott, G.D. 1995. The relationship between bitumens and mineralization in the South Pennine Orefield, central England. Journal of the Geological Society, London, 152, 751–765. <https://doi.org/10.1144/gsjgs.152.5.0751>

Author’s response: This reference is now included, the section of text now reads...(Ewbank et al., 1995; Fraser and Gawthorpe, 1990; Leeder, 1982, 1988).

Reviewer 3

Main comments

1. The paper uses two cored Bowland Shale data points from the northern England Carboniferous province namely Grange Hill in the Bowland Basin and Rempstone -1 in the Widmenpoo; Gulf. The boreholes are roughly 200kms apart and have been chosen to compare similar shales at different maturity levels. They are also different ages and this is not discussed. They also have potentially different provenance areas; Grange Hill from the north east and Rempstone from the south. Probably not that relevant to the research but should be mentioned.

Author’s response: To cover this omission concerning the geological age of the shales. We have added the following to the Methods section (Bowland Shale description).

Whereas the Grange Hill-1 3113m sample is from the Lower Bowland Shale (Brigantian, Dinantian) with a provenance from the prodelta sources to the north east, the Rempstone-1 665-667m sample comes from the Upper Bowland Shale (Pendleian, Namurian) with a provenance from the prodeltas to the north and south of the Widmerpool Trough on the Derbyshire and Midlands Highs.

2. My understanding of the maturity of the shales at Rempstone differs somewhat from the authors. My analyses indicate these are early mature for oil generation. In fact it is difficult to explain the conventional Rempstone oilfield interbedded with the Bowland Shale in the absence a mature local source rock. This observation would not make a big difference to the analysis and subsequent conclusions.

Author’s response: We agree that the Rempstone shale can be considered as “oil window mature” and this is used throughout the text, as further indicated in the responses to the text annotations below.

3. The detailed analysis of the methods used could be in my view shortened and replaced by some key information on the resource calculation and estimates currently consigned to supplementary information.

Author’s response: As suggested, we have moved the description of the resource estimation method for the entire Bowland shale to the main article as an extra section under Methods. However, we consider it is essential to keep the full descriptions of the other experimental procedures in the Methods to enable our researchers to replicate these, particularly the high pressure water pyrolysis

Text annotations

Comment 1, Line 20: Insert "on well bore core sample" after experiments.

Author's response: Inserted.

Comment 2, Line 21: Why uplifted basins specifically.

Author's response: Shale gas basins are all onshore and therefore uplifted relative to sea level.

Comment 3, Lines 25-26: The maximum GIP is vastly lower than previously estimated, where does this come?

Author's response: Andrews (2013) is not cited since references are not allowed in the abstract. The last sentence in the introductory has been modified to read "Extrapolating our findings to the whole Bowland Shale, then the maximum GIP equate to potentially economically recoverable reserves of less than 10 years of current UK gas consumption"

Comment 4, 27-30: It's not only HPWP that is responsible for the difference. Poor understanding of the basin history and stratigraphy of the source rocks is also important.

Author's response: This comment also relates to comment 3 above, and the statements have been modified to read "The gas yields from HPWP of UK Bowland shales are comparable with those from degassed cores, with the ca. 1% porosity being sufficient to accommodate the gas generated. Extrapolating our findings to the whole Bowland Shale, then the maximum GIP equate to potentially economically recoverable reserves of less than 10 years of current UK gas consumption".

Comment 5, line 43: Contrary to the evidence for US shale

Author's response: Line 43 modified to read "Contrary to US producing shales (Barnett, Marcellus and Fayetteville) having VR>1.4% Ro."

Comment 6, line 44: An assumption that all Lower Carboniferous shales were potential source rocks.

Author's response: Text modified to "The high estimate obtained by Andrews (2013) was also due to the assumption that all Carboniferous shales of the Bowland basin are potential shale gas source rocks have been added as suggested by the reviewer."

Comment 7, line 59: My understanding of the maturity of the shales at Rempstone differs somewhat from the authors. My analyses indicate these are early mature for oil generation. In fact it is difficult to explain the conventional Rempstone oilfield interbedded with the Bowland Shale in the absence of a mature local source rock. This observation would not make a big difference to the analysis and subsequent conclusions.

Author's response: The above comment is same as in lines 75, 246 and 278. The authors are in complete agreement with the reviewer. The word "immature" for the Rempstone source rock have been replaced with "oil-window mature" throughout the manuscript.

Comment 8, lines 59-71: Different basins, different structures (??), different ages of possible organic rich shales"

Author's response: "It must be remembered that some differences between the different studies arise from the samples coming from different locations within the basin with consequent differences in stratigraphical, structural and tectonic histories of the different parts of the same basin" has been added.

Comment 9, line 67: Delete then and insert on the basin after considered

Author's response: Change made.

Comment 10, lines 79-80: Insert supplementary information after 300 bar.

Author's response: "(Supplementary Table S2)" added.

Comment 11, line 104: Insert generation between direct and from.

Author's response: Change made.

Comment 12, lines 219-223: "TCF normal term used by industry".

Author's response: tscf have been replaced with TCF all throughout the manuscript.

Comment 13, bottom of page below line 229, reads like "we can argue about original carbon content of the Bowland shale formation".

Author's response: We consider the uncertainty over our global estimate of the organic carbon content and the h=variability expected are addressed fully in our description of the shale in the Methods section.

Comment 14, line 231: Optimistic

Author's response: we agree with the reviewers opinion but we have not inserted "optimistic" because it is view which is difficult to reference.

Comment 15, line 246: Immature.

Author's response: Already addressed above (comment 8).

Comment 16, general comment on Fig 8: Comment includes labelling Grange location, correcting the scale at the bottom of the figure, and producing a high resolution image etc.

Author's response: Figure 8 was removed because it is not essential and we could not make these corrections from the earlier reference.

Comment 17, line 278: Immature.

Author's response: Already addressed above (comment 7).

Comment 18, line 292: Insert gas mature.

Author's response: "Gas mature shale "inserted.

Comment 19, lines 361 and 363: Shales change to shale

Author's response: change made.